# Genome-wide chromatin contacts of super-enhancer-associated lncRNA identify LINC01013 as a regulator of fibrosis in the aortic valve

**Arnaud Chignon[1], Déborah Argaud[1], Marie-Chloé Boulanger[1], Ghada Mkannez[1], Valentin Bon-Baret[1], Zhonglin Li[1], Sébastien Thériault[2], Yohan Bossé[3], Patrick Mathieu[1] ***

**1** Laboratory of Cardiovascular Pathobiology, Quebec Heart and Lung Institute/Research Center, Department of Surgery, Laval University, Quebec, Canada, **2** Department of Molecular Biology, Medical Biochemistry and Pathology, Laval University, Quebec, Canada, **3** Department of Molecular Medicine, Laval University, Quebec, Canada

* patrick.mathieu@fmed.ulaval.ca

## Abstract

Calcific aortic valve disease (CAVD) is characterized by a fibrocalcific process. The regulatory mechanisms that drive the fibrotic response in the aortic valve (AV) are poorly understood. Long noncoding RNAs derived from super-enhancers (lncRNA-SE) control gene expression and cell fate. Herein, multidimensional profiling including chromatin immunoprecipitation and sequencing, transposase-accessible chromatin sequencing, genome-wide 3D chromatin contacts of enhancer-promoter identified *LINC01013* as an overexpressed lncRNA-SE during CAVD. *LINC01013* is within a loop anchor, which has contact with the promoter of *CCN2 (CTGF)* located at ~180 kb upstream. Investigation showed that *LINC01013* acts as a decoy factor for the negative transcription elongation factor E (NELF-E), whereby it controls the expression of *CCN2*. *LINC01013-CCN2* is part of a transforming growth factor beta 1 (TGFB1) network and exerts a control over fibrogenesis. These findings illustrate a novel mechanism whereby a dysregulated lncRNA-SE controls, through a looping process, the expression of CCN2 and fibrogenesis of the AV.

## Author summary

Calcific aortic valve disease is the most common heart valve disorder characterized by a thickening of the aortic valve resulting from fibrotic and calcific processes. Because the aortic valve replacement is currently the only therapeutic option, the identification of key molecular processes that control the progression of the disease could lead to the development of novel noninvasive therapies. Growing evidence suggests that long noncoding RNAs (lncRNAs) fine tune gene expression in health and disease states. By using a multi-dimensional profiling including genome-wide 3D enhancer-promoter looping data, we identified *LINC01013*, a lncRNA, as a regulator of fibrogenesis. Specifically, we found that

**Data Availability Statement:** Data are available in GEO (accession numbers GSE154510, GSE154511 and GSE154512).

**Funding:** This work was supported by the Canadian Institutes of Health Research grants to P. M. (FRN148778, FRN159697) and the Quebec Heart and Lung Institute Fund. Y.B. holds a Canada Research Chair in Genomics of Heart and Lung Diseases. S.T. holds a junior scholarship from Fonds de Recherche du Québec-Santé (FRQS). P. M. holds a Fonds de Recherche du Québec-Santé (FRQS) Research Chair on the Pathobiology of Calcific Aortic Valve Disease. The funders had no role in study design, data collection and analysis, decision to publish, or preparation of the manuscript.

**Competing interests:** The authors have declared that no competing interests exist.

*LINC01013* is located in a cluster of distant enhancers (super-enhancer) in aortic valve interstitial cells and has significant long-range looping with the promoter of *CCN2*, a gene that orchestrates fibrogenesis. We discovered that *LINC01013* is acting as a decoy factor for a negative transcription elongation factor, whereby it controls the transcription of *CCN2*. In turn, higher expression of *LINC01013* during calcific aortic valve disease promoted the expression of CCN2 and a fibrogenic program. These findings provide evidence that *LINC01013* is a key regulator of fibrogenesis in CAVD.

## Introduction

Calcific aortic valve disease (CAVD) is the most frequent heart valve disorder [1]. The progression of CAVD culminates in aortic valve (AV) stenosis, which can be treated by aortic valve replacement (AVR) in symptomatic patients. AVR either surgical or by percutaneous approach is associated with elevated cost and significant morbidity/mortality [2]. Hence, the identification of key underpinning processes that control the development of CAVD could lead to medical therapy to prevent the progression of CAVD. Analyses of surgically explanted AVs have shown that the thickening of leaflets, a hallmark feature of CAVD, is associated with an excess of extracellular matrix components such as collagen fibrils [2]. Research conducted over the last several years has underlined several molecular processes involved in the development of CAVD and has highlighted that valve interstitial cells (VICs) have a high plasticity [3]. As such, according to different cues VICs may transition from quiescent to activated cells with an increased potential for fibrogenesis [3]. Studies have consistently shown that the transforming growth factor (TGF) beta pathway was involved in fibrogenesis and the development of CAVD [4]. Pre-clinical and clinical data have highlighted the benefit of targeting the TGF beta pathway to reduce the cardiac fibrosis [5] and the key role of the TGF beta-CCN2/CTGF axis [6]. However, key regulators of the TGF beta pathway that orchestrate and instruct a fibrogenic programme in the AV are still largely unknown.

Transcription factors and enhancers control gene expression pattern [7]. Super-enhancers (SE) are groups of distant-acting *cis*-regulatory elements (enhancers) in close proximity, which are enriched in chromatin loops with the promoter of genes [8]. The folding pattern of the chromatin brings distant elements, such as SE, in contact with gene promoters and exert a control on the transcriptional process. Studies have highlighted that SE are cell-specific and control cell fate and identity [8]. Enhancer and SE may express noncoding RNA including long noncoding RNA (lncRNA) [9,10]. The expression of lncRNA derived from SE (lncRNA-SE) provides another layer of control on gene expression [10]. Studies on cardiac fibrosis have demonstrated that lncRNA or lncRNA-SE could be targeted to reduce cardiac fibrosis [11,12]. In this work, we hypothesized that dysregulated lncRNA-SE expressed by VICs may contribute to fibrogenesis, an early and key process in the development of CAVD. By using functional assays and genomic multidimensional profiling of VICs and AVs (S1 Fig), we identified *LINC01013*, a CAVD-dysregulated lncRNA-SE within a regulatory DNA loop that controls the transcriptional process of gene enriched in the TGF beta pathway and promotes fibrogenesis.

## Results

### Super-enhancer in valve interstitial cells

SEs exhibit a strong enrichment for the active H3K27ac regulatory mark and play a major role in the signaling pathways. We thus wondered if SEs could be involved in the biology of the

AV. For that purpose, in VICs, we first generated chromatin immunoprecipitation for
H3K27ac followed by sequencing to identify active regulatory elements. We identified 82,662
strong peaks corresponding to putative regulatory elements. We next used these data to iden-
tify SEs by using the ranking of super-enhancer (ROSE) algorithm as described previously
[13,14]. ROSE is the most utilized approach to identify SEs; briefly the algorithm performs a
ranking of strong and close H3K27ac peaks, and then identifies SEs from peaks plotted on a
scatter plot upper an inflection point for which the slope is 1. Thus, in VICs, we identified
1085 SEs (S1 Table). We wanted to replicate these data by using another algorithm integrated
into the HOMER tools, and we observed that 83.3% of the SEs identified by HOMER (S2
Table) are also found by the ROSE algorithm. By using GREAT (see the Methods section), we
found that the 1085 SEs were highly enriched in gene ontology (GO) for extracellular matrix
organization (GO:0030198) (P = 2.01 x $10^{-21}$) (Fig 1A). Active regulatory elements like
enhancers and SEs are characterized by a chromatin accessibility. In VICs, we performed an
assay for transposase-accessible chromatin and massively parallel sequencing (ATAC-seq) to
capture the open chromatin regions. We found that there was a significant and a positive cor-
relation of the ATAC-seq signal close to the SEs (observed/expected ratio = 5.2, P<1 x $10^{-16}$
binomial test) (Fig 1B): these findings attest the importance of the SEs identified as active regu-
latory elements. By using an extensive annotation of lncRNAs (GENCODE [15] version 32)
that was intersected with the SE dataset, we identified 324 SEs from which at least one lncRNA
is transcribed, called lncRNA-SE (S3 Table), which represent 30% of all the SE in VICs (Fig
1C). Based on the ROSE algorithm [8], lncRNA-SE were ranked higher than regular SE
(R-SE), from which no lncRNA is transcribed (P<0.0001, Wilcoxon ranked-sum test) (Fig
1D). We thus wondered whether lncRNA-SE were associated with increased open chromatin
compared to R-SE. In ATAC-seq, the tag density for the core region (±5kB) was significantly
increased in lncRNA-SE compared to R-SE (P<0.0001, Wilcoxon ranked-sum test) (Fig 1E
and 1F). SE are enriched in 3D chromatin looping involved in gene regulation [14]. In VICs,
genome-wide enhancer-promoter 3D mapping was evaluated by using H3K27ac HiChIP,
which generated 454,964,884 unique paired-end tags. Fig 2A shows an example of an interac-
tion matrix for the chromosome 6 with a resolution of up to 1kb. S2 Fig shows interaction
matrices for the others chromosome. In total, in VICs we mapped 36,229 high-confidence
loops at false discovery rate (FDR)<0.01. The mean number of protein coding gene promoters
(±1 kb from transcription start site) mapped by 3D chromatin looping was 1.2 and 1.9 for
R-SE and lncRNA-SE respectively. Taken together, these data highlight that VIC-associated
lncRNA-SE are highly ranked SE with increased open chromatin and are spatially coordinated
with distant protein coding genes.

## Differentially expressed lncRNA-SE and 3D genome mapping

We next wondered if lncRNA-SE were differentially expressed during the development of
CAVD. By using RNA-sequencing (RNA-seq) in 10 control nonmineralized AVs and 19 surgi-
cally-explanted mineralized AVs (CAVD), we profiled the expression of lncRNAs. RNA
sequencing was annotated with GENCODE version 32. Fig 2B presents a heatmap of differen-
tially expressed lncRNAs between control nonmineralized and mineralized AVs (CAVD).
When comparing CAVD to control AVs, 178 and 111 lncRNAs were upregulated and downre-
gulated respectively (FDR<0.05) (S4 Table). Among the differentially regulated lncRNAs
there were 18 lncRNA-SE (10 upregulated and 8 downregulated in CAVD) (S5 Table). In
order to identify the putative genes regulated by lncRNA-SE we mapped these regulatory ele-
ments to their protein coding genes (±1 kb from the transcription start site) by using chroma-
tin looping information obtained with the H3K27ac HiChIP in VICs. Among the 18

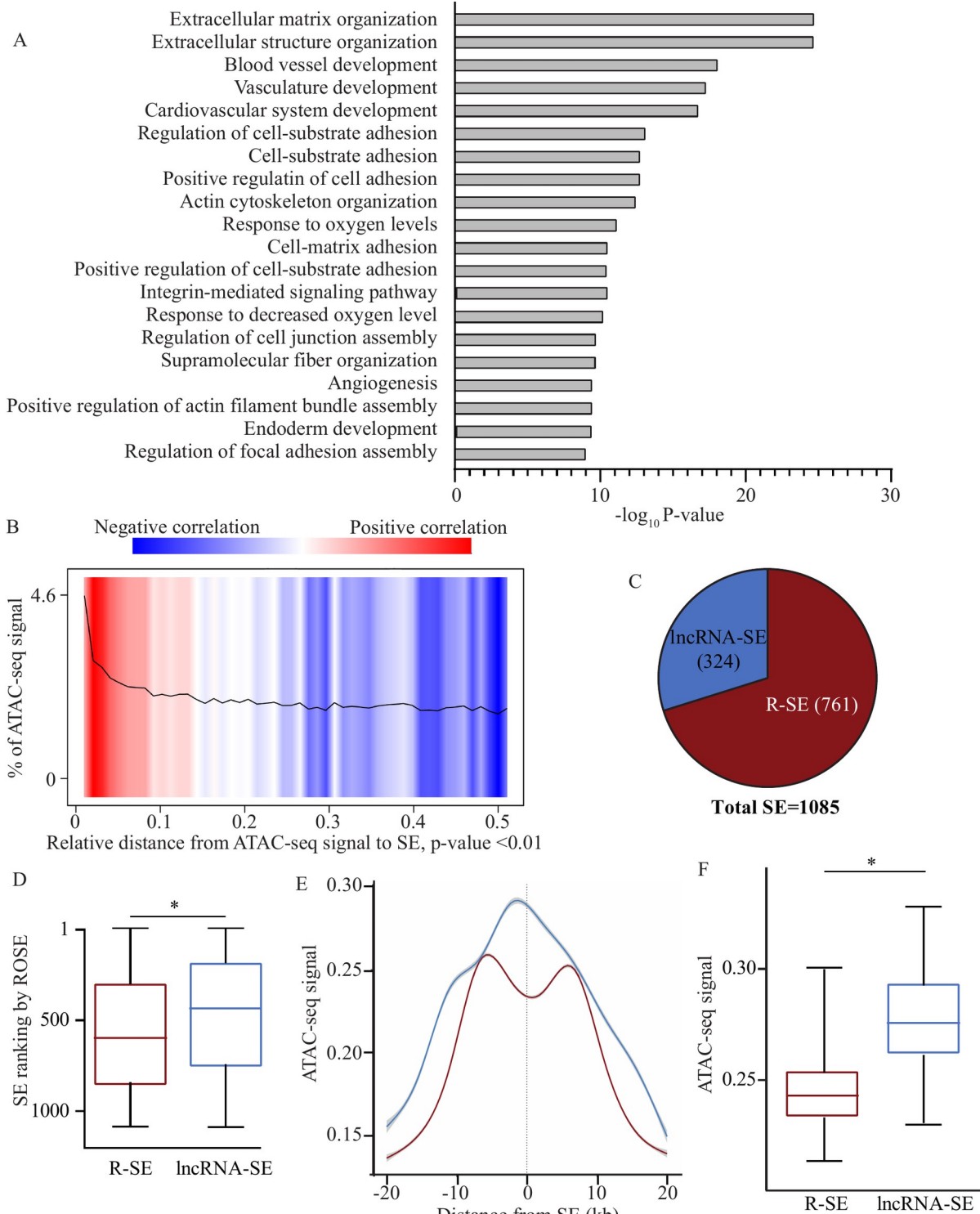

**Fig 1. Super-enhancer (SE) in valve interstitial cells (VICs).** A) Gene ontology enrichment for SE in VICs (adjusted P-value). B) ATAC-seq signal relative to distance from SE. C) Pie chart showing that 30% of VICs SE intersect with lncRNA (lncRNA-SE); SEs for which no lncRNA is transcribed are defined as regular SE (R-SE). D) The ranking of super-enhancer (ROSE) algorithm indicates that lncRNA-SE rank higher than R-SE in VICs. E) Signal intensity for open chromatin (ATAC-seq) was higher at the core of lncRNA-SE compared with R-SE. F) Box plot revealing higher ATAC-seq signal for lncRNA-SE.

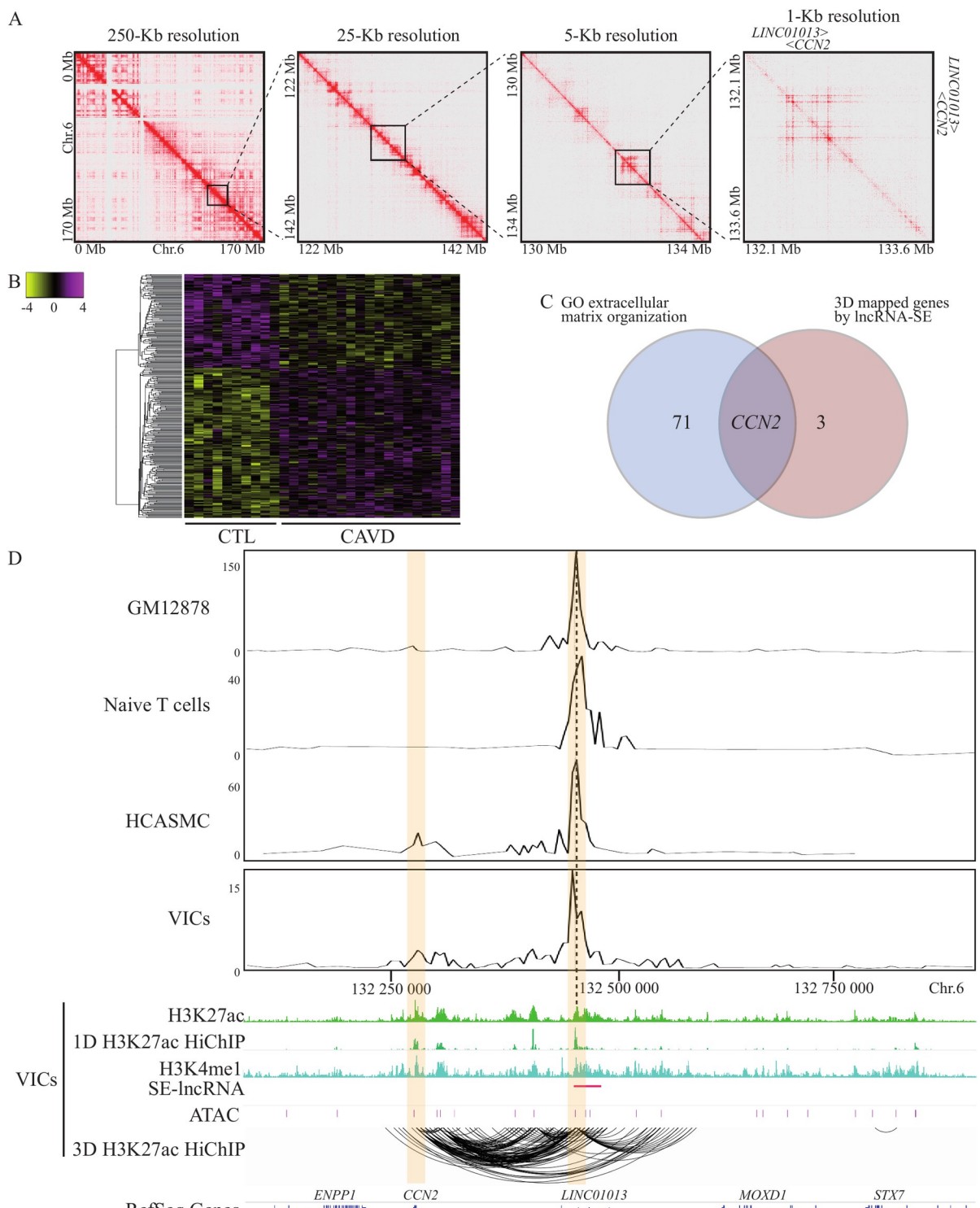

**Fig 2. CAVD associated lncRNA-SE and 3D mapping.** A) Chromosome 6 interaction matrices at the indicated resolutions from H3K27ac HiChIP in VICs. B) Heat map of lncRNA expression in 10 control and 19 calcified aortic valves. C) Venn diagram of genes mapped by chromatin contact with differentially expressed lncRNA and genes encompassed in the GO for extracellular matrix organization displaying that only *CCN2* is common to both sets. D) Virtual 4C representations of chromatin interactions at 6q23.2 revealing cell line specific contact between lncRNA-SE, *LINC01013* and *CCN2* promoter region. Tracks for H3K27ac ChIP-seq and HiChIP (1D), H3K4me1 ChIP-seq, SE-lncRNA, ATAC-seq signal and H3k27ac HiChIP (3D).

differentially expressed lncRNA-SE, we found that in total 4 were having significant chromatin contacts with distant promoters of protein coding genes (S6 Table). The mean distance between a differentially expressed lncRNA-SE and a protein coding gene was ~89 kb. We wondered whether genes mapped by H3K27ac HiChIP and having loops with lncRNA-SE could be part of the GO for extracellular matrix organization, the highest enriched ontology for SE in VICs. Fig 2C shows a Venn diagram of genes mapped by chromatin contact with differentially expressed lncRNA-SE and genes encompassed in the GO for extracellular matrix organization. One gene promoter for *CCN2* (6q23.2) having loops with a differentially expressed lncRNA-SE overlapped with the GO for extracellular matrix organization. We next examined the genome 3D interactions at 6q23.2 by using virtual 4C (v4C), which provides an anchor as a viewpoint of all interactions (with the anchor) within a genomic region and visualized in 2D. Fig 2D shows v4C representation of chromatin interactions at 6q23.2 in VICs where *LINC01013/ LOC100507254*, a poorly conserved lncRNA-SE according to the NONCODE database [16], interacts with the promoter region of *CCN2* located ~180 kb upstream. High confidence significant loops identified strong chromatin contacts between the promoter of *LINC01013* and the promoter of *CCN2* as well as many significant non-coding chromatin contacts at these loci. *LINC01013* is overexpressed in mineralized compared to control nonmineralized AVs and SE region for *LINC01013* extends over ~30 kb. *CCN2* encodes for the connective tissue growth factor (CTGF), a matricellular protein involved in cell adhesion, proliferation and synthesis of extracellular matrix [17,18]. Interrogation of enhancer-promoter chromatin contacts using H3K27ac HiChIP at the 6q23.2 locus showed that similarly to VICs, *LINC01013* interacts with *CCN2* in human coronary artery smooth muscle cells (HCASMC), whereas there is no chromatin looping at this locus in blood cells such as naive T cells and GM12878 (B cell line) (Fig 2D). These data underline that spatial organization of chromatin at the *LINC01013* locus is likely specific to mesenchymally-derived cells. The S3 Fig gives a summary of the sequential analysis which conducted to the prioritization of *LINC01013*.

### Characterization of the *LINC01013* locus

We confirmed with a higher resolution the data obtained in HiChIP by using chromatin conformation capture (3C) and pairs of primers covering the region of *LINC01013* (S7 Table). In VICs, the 3C assay showed that the promoter region of *CCN2* (used as an anchor) interacted with the promoter region of *LINC01013* (Fig 3A). Alternative splicing of *LINC01013* produces transcripts with 2 or 3 exons. In VICs, only the 3 exons transcript was expressed, whereas the 2 exons transcript was not detected by RT-qPCR (Fig 3B). We generated data to confirm the RNA-seq analysis. We measured the level of transcripts encoding for *LINC01013* in 18 control nonmineralized and 26 mineralized AVs by using quantitative RT-qPCR. Table 1 presents the demographic data for the patients for whom the AVs were obtained. We found that *LINC01013* was increased by 2.95-fold in mineralized compared to control AVs (P<0.0001; age- and sex-adjusted P = 0.0003) (Fig 3C). The noncoding potential of *LINC01013* was confirmed by using CPAT [19] (coding probability = 0.04), a logistic regression algorithm to probe the coding potential (S8 Table). In VICs, a ChIP-seq with an antibody directed against H3K4me1, a histone mark of enhancer, showed strong peaks over the *LINC01013* region, which overlapped with H3K27ac and open chromatin (Fig 2D). We next asked whether the dysregulation of *LINC01013* in CAVD could be linked to a disease-associated epigenetic process. We measured by quantitative ChIP the enrichment of H3K27ac at the promoter of *LINC01013* in control and mineralized AVs. Compared to control AVs, we found that the H3K27ac mark at the promoter of *LINC01013* was increased by 5.41-fold in mineralized AVs

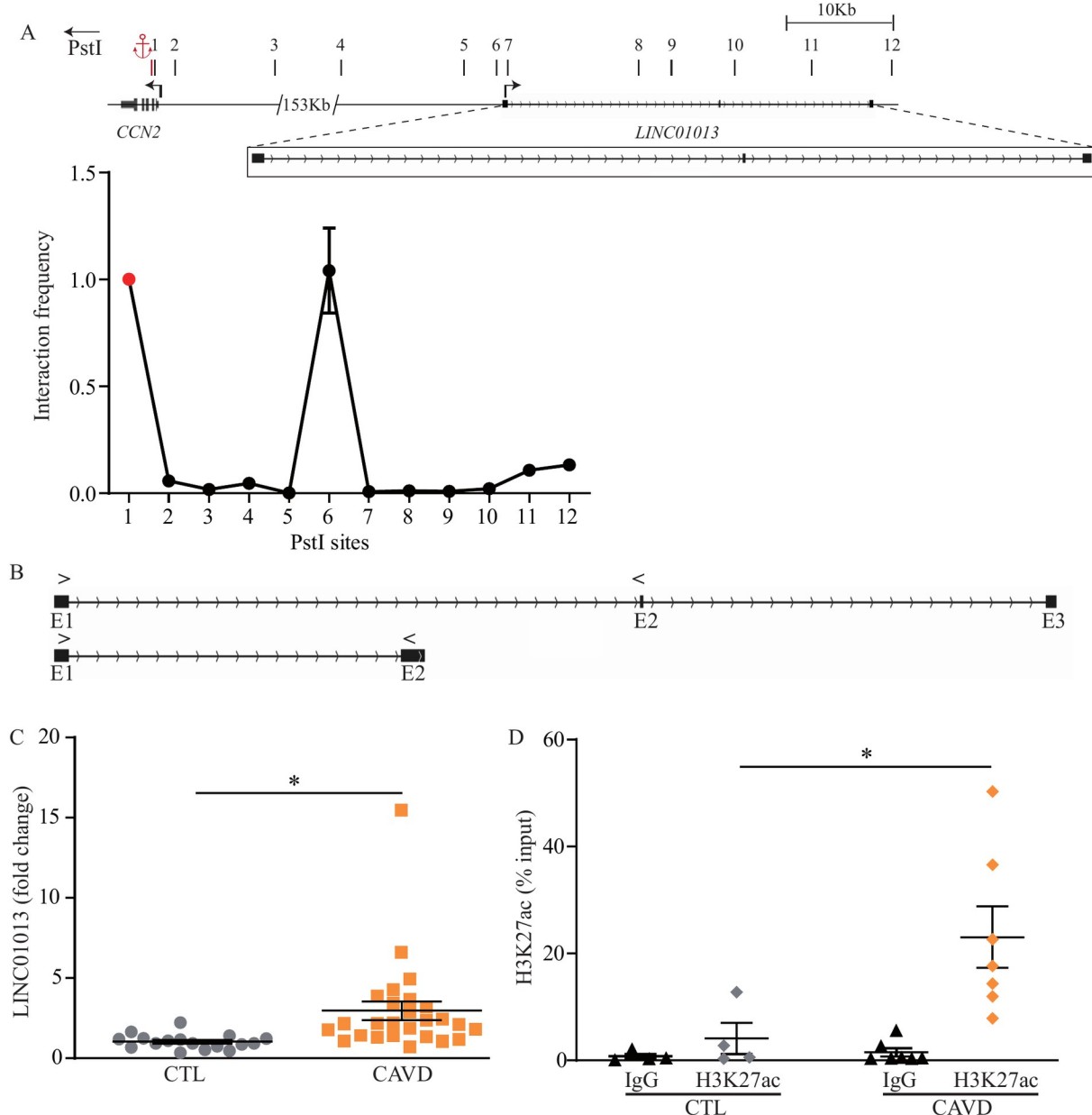

**Fig 3. LINC01013 locus characterization.** A) Chromatin conformation capture (3C) assay demonstrating contact between *LINC01013* and *CCN2* promoter regions (n = 4). B) Quantitative PCR (qPCR) primers design represented by the arrows to amplify specifically the long or the short transcripts of *LINC01013* (3 or 2 exons). C) qPCR assay confirming increased RNA expression of *LINC01013* in mineralized (n = 27) compared to non-mineralized (n = 17) aortic valves. D) ChIP-qPCR indicating higher H3K27ac signal in calcified valves at the *LINC01013* promoter (CTL: n = 4, CAVD: n = 7).

(Fig 3D). Taken together, these orthogonal data suggest that *LINC01013* is located in extended histone marks of active regulatory elements and is spatially coordinated with *CCN2*.

## LINC01013 and transcriptional response at CCN2

We assessed the expression of *LINC01013* in the cytosol and the nuclear compartments of VICs. After cell fractionation, we observed that *LINC01013* was expressed in both the nuclear

**Table 1. Clinical data for control and CAVD.**

| | Control valves | CAVD | P-value |
|---|---|---|---|
| | (n = 17) | (n = 27) | |
| Age | 50 ± 3,71 | 68 ± 1,53 | 0,0001 |
| Male (%) | 88 | 63 | 0,0897 |
| Smoking (%) | 12 | 22 | 0,5149 |
| Hypertension (%) | 41 | 81 | 0,0094 |
| Diabetes (%) | 0,0 | 0,0 | 1 |
| Coronary heart disease (%) | 17,6 | 18,5 | 1 |
| Bicuspid Avs (%) | 0,0 | 37,0 | 0,0037 |
| BMI (kg/m2) | 25,8 ± 1,2 | 29,7 ± 1,2 | 0,0135 |
| Waist circumference (cm2) | 114 ± 2,06 | 105,5 ± 3,8 | 0,8213 |
| Statins (%) | 47 | 70 | 0,1053 |
| AV area (cm2) | - | 1 ± 0,1 | - |
| Aortic peak gradient (mmHg) | - | 63,0 ± 5,5 | - |
| Aortic mean gradient (mmHg) | - | 37,1 ± 3,4 | - |
| Triglycerides (mmol/L) | 1,65 ± 0,27 | 1,16 ± 0,08 | 0,9267 |
| LDL (mmol/L) | 1,93 ± 0,25 | 2,08 ± 0,13 | 0,3215 |
| HDL (mmol/L) | 1,04 ± 0,09 | 1,35 ± 0,09 | 0,0144 |
| Creatinine (μmol/L) | 121,94 ± 8,96 | 79,96 ± 4,65 | 0,9998 |
| Creatinine clearance (ml/min) | 65,05 ± 5,31 | 77,75 ± 5,71 | 0,0581 |

and cytosol fractions (Fig 4A). To explore the functional role, we designed an antisense oligo-nucleotide (ASO) targeting *LINC01013*. In VICs, the transfection of ASO provided a strong reduction of *LINC01013* (Fig 4B). Transfection of the ASO against *LINC01013* in VICs also led to a significant reduction of transcripts encoding for *CCN2* measured after 72 hours (Fig 4C). In the same line, the knockdown of *LINC01013* was followed by a significant reduction of CCN2 (CTGF) at the protein level (Fig 4D). Promoter-proximal pausing of RNA polymerase II (RNAPII) is a widespread regulatory mechanism of the transcriptional process [20]. NELF (negative elongation factor) is a protein complex interacting with the RNAPII [21] which restrains its progression usually 20 to 60 bp downstream from the transcription start site (TSS) [20]. The recruitment of the NELF complex is associated with the phosphorylation on serine 5 of the c-terminal domain of RNAPII (RNAPII-Ser5p) enriched at paused promoters [20,22]. Previous studies have identified that some noncoding RNAs can modulate the recruitment of the NELF complex by sequestering the NELF-E subunit [23,24]. Considering that *LINC01013* has a nuclear localization and its locus is spatially coordinated with the promoter of *CCN2*, we hypothesized that *LINC01013* can act as a decoy factor for the NELF-E subunit and thus modulate its recruitment at *CCN2*. We performed a RNA immunoprecipitation (RIP) assay to evaluate the possible interaction between *LINC01013* and NELF-E. In VICs, a pull-down of the NELF-E subunit followed by a quantitative RT-qPCR showed that *LINC01013* was enriched by 6-fold compared to the IgG control (Fig 4E). In addition, quantitative ChIP demonstrated that the knockdown of *LINC01013* in VICs led to a significant enrichment of the NELF-E subunit at the TSS of *CCN2* (Fig 4F). However, at the promoter of *LINC01013*, we observed a poor enrichment of NELF-E compared to the IgG control and no impact on the recruitment after the knockdown (S4A Fig). In line with these findings, we found that the knockdown of *LINC01013* was accompanied by a higher level of RNAPII-Ser5p at the TSS of *CCN2* (Fig 4G). However, total RNAPII revealed a similar occupancy at the TSS of *CCN2* with or without the knockdown (Fig 4H). These data thus indicate that *LINC01013* controls the magnitude of the

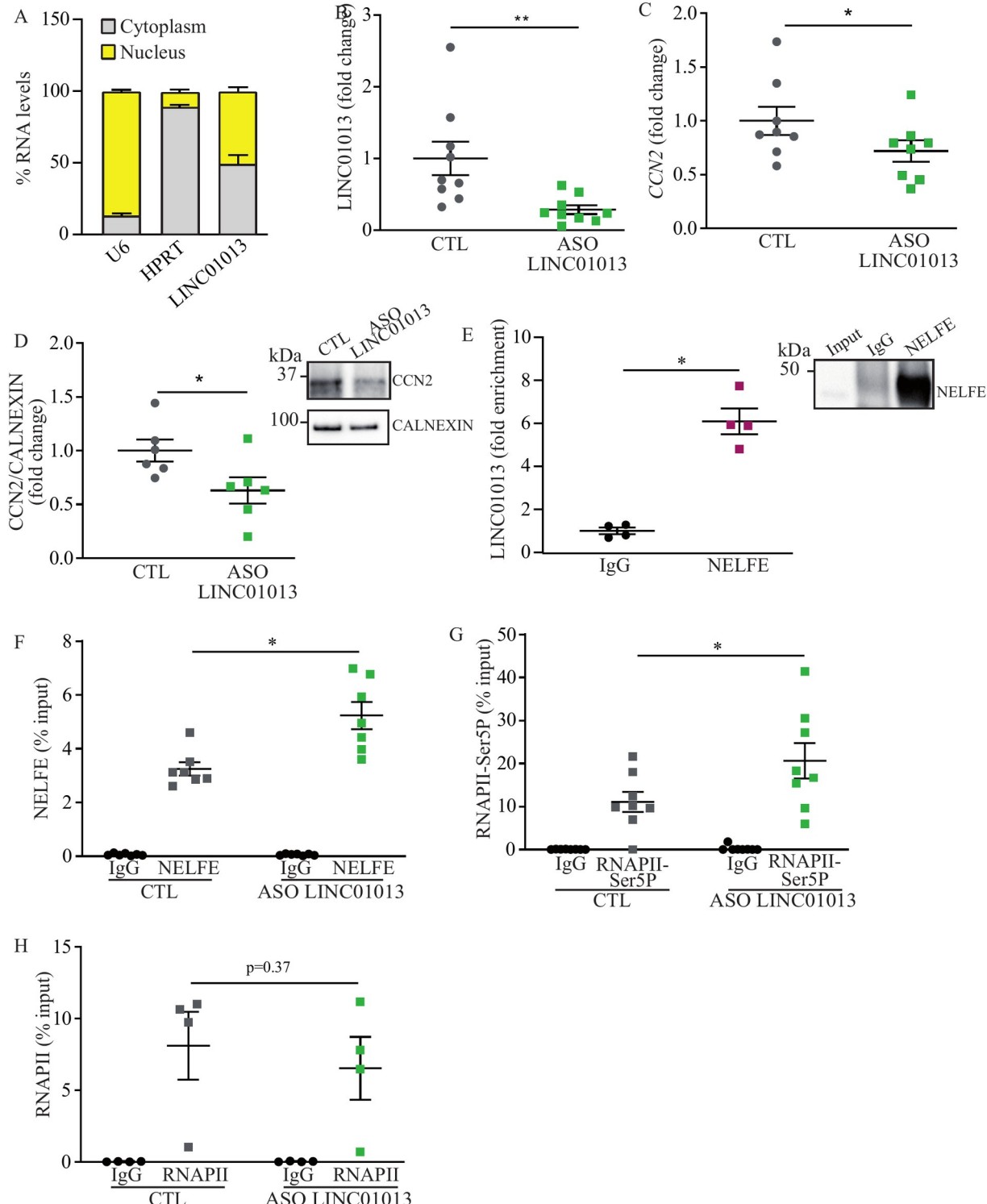

**Fig 4. *LINC01013* regulates the transcription at *CCN2*.** A) Cell fractionation and qPCR showing *LINC01013* expression in both the nuclear and cytosolic compartments (n = 3). B-D) *LINC01013* (n = 9) (B) and *CCN2* (n = 8) (C) RNA levels and CCN2 protein (n = 6) (D) decrease following treatment with *LINC01013* ASO. E) RNA immunoprecipitation assay (RIP) indicating an interaction between the *LINC01013* and NELF-E protein, western blot demonstrating NELF-E immunoprecipitation (n = 4). F) ChIP-qPCR revealing a rise in NELF-E protein at the *CCN2* promoter after treatment with *LINC01013* ASO (n = 7). G-H) ChIP-qPCR showing increased RNAPII-Ser5P (n = 8) (G) occupancy at the *CCN2* promoter after treatment with the ASO against *LINC01013*, but unchanged total RNAPII (n = 4) (H).

transcriptional response at *CCN2* through a modulation of RNAPPII pausing by acting as a decoy factor for the NELF-E subunit.

## TGF beta pathway and LINC01013

We extracted a protein-protein interaction (PPI) network from RNA-seq in calcified AVs in order to evaluate the interaction of CCN2 and its pathway. For that purpose, we used an experimentally validated dataset including 156,317 PPIs. The extracted network in CAVD included 446 nodes (gene derived proteins) and 591 edges (connections) and was enriched for TGF beta signaling pathway ($P_{adjusted} = 4.17 \times 10^{-72}$) (Fig 5A and 5B). Highly connected nodes such as SMAD2, SMAD3, TGFB1 and COL1A1 were linked to CCN2. We measured in VICs the expression of COL1A1, a widely expressed collagen in the AV, in response to a knockdown of *LINC01013* knowing its impact on the expression of CCN2. In VICs, the knockdown of *LINC01013* lowered the protein level of COL1A1 by 55% after 72 hours (Fig 6A). Also, the transfection of an ASO targeting *LINC01013* in VIC cultures lowered the extracellular secretion of pro-collagen 1-α by 31% (Fig 6B). Immunostaining of COL1A1 in VICs in culture demonstrated similar results on whole cells (S4B Fig). TGF beta is an enriched pathway in CAVD and an important promoter of VIC activation [3]. We tested whether TGFB1 may impact on the expression of *CCN2* through the activation of *LINC01013*. First, in TGFB1-treated VIC cultures (10 ng/ml), we found that the expression of *LINC01013* was increased by 1.9-fold (Fig 6C). Similarly, the expression of *CCN2* was increased by 3.1-fold in TGFB1-treated VIC cultures (Fig 6D). Of interest, the ASO against *LINC01013* abrogated the TGFB1-induced rise of *CCN2* (Fig 6D). Also, in western blot, the level of CCN2 increased following a treatment with TGFB1, whereas the knockdown of *LINC01013* largely abrogated this rise (Fig 6E). We next treated VIC cultures with TGFB1 and we measured the level of collagen with and without a knockdown of *LINC01013*. The knockdown of *LINC01013* in VIC cultures prevented the TGFB1-induced rise of *COL1A1* transcript and the secretion of pro-collagen type 1-α (Fig 6F and 6G). Data obtained from an immunostaining of COL1A1 in whole cells are consistent with these results (S4 Fig). Similarly, the knockdown of *CCN2* by an ASO, which significantly reduced the transcript level of the target, also abrogated TGFB1-induced rise of *COL1A1* (Fig 6H and 6I). Taken together, these data obtained in VICs indicate that TGFB1-mediated expression of collagen relies on a *LINC01013-CCN2* pathway.

## Discussion

Herein, genomic profiling of VICs identified that SE are enriched for extracellular matrix organization. Among these clusters of regulatory elements, lncRNA-SE are enriched in open chromatin and loops with gene promoters. We identified a lncRNA-SE, *LINC01013*, which is upregulated during CAVD. Functional assessment showed that *LINC01013* is within a DNA loop that has contacts with the promoter of *CCN2*. *LINC01013* is acting as a decoy factor for NELF-E, whereby it controls the transcription at *CCN2*. Increased expression of *LINC01013* promotes the dissociation of NELF-E from chromatin and the expression of *CCN2*, which encodes for a matricellular protein that stimulates fibrogenesis (Fig 6J).

### LncRNA and super-enhancers in VICs

In CAVD, previous studies have identified dysregulated lncRNAs such as TUG1 [25] and MALAT1 [26], which act as sponges of miR-204, a microRNA that regulates the level of the osteogenic transcription factor *RUNX2*. Also, the expression of the lncRNA H19 is increased in VICs during CAVD and controls the expression of *NOTCH1*, a key regulator of cell fate [27]. In the heart, CARMEN is a lncRNA-SE, which regulates cardiac specification of precursor cells [28]. To our knowledge, the present work is the first to identify and characterize

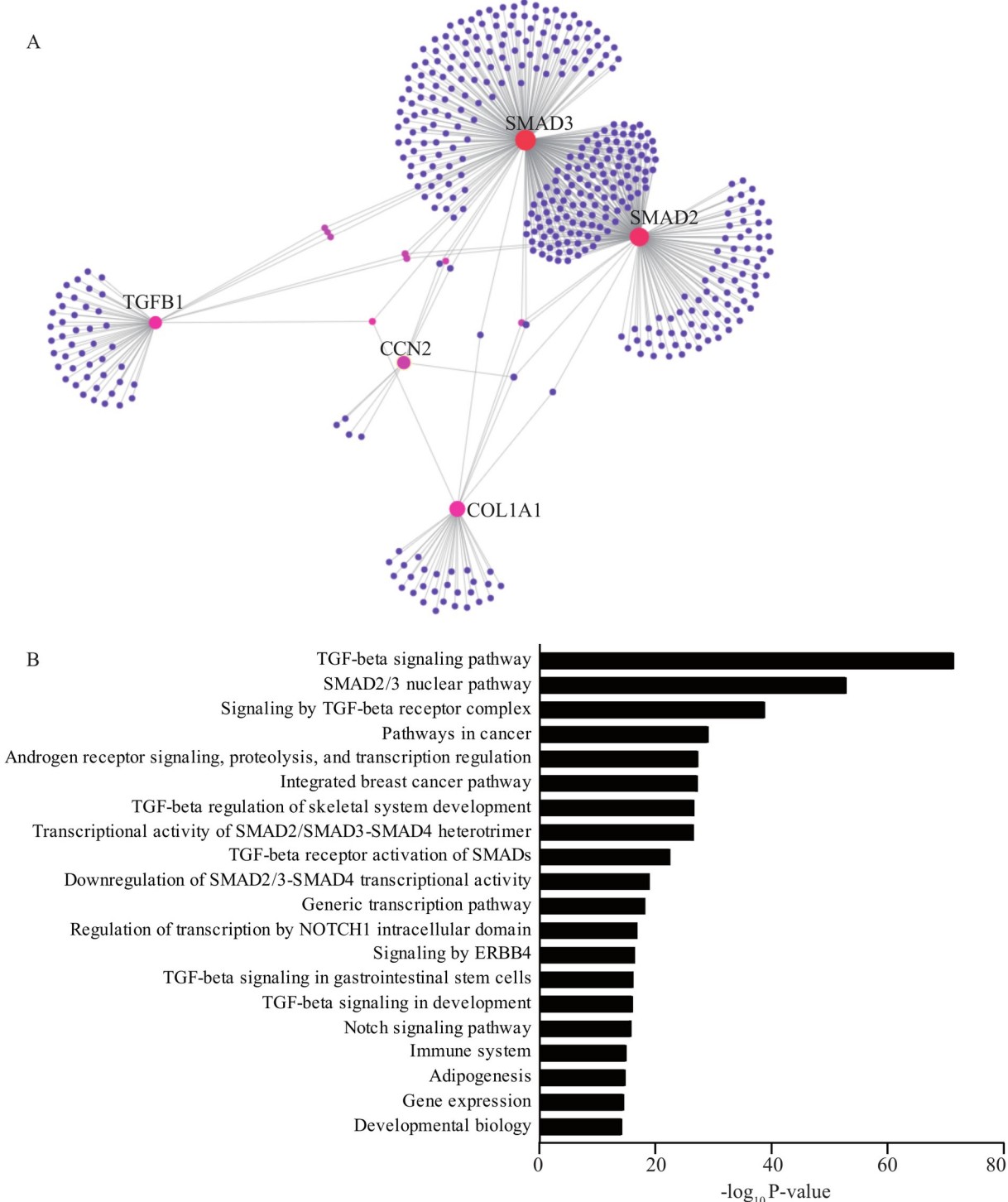

**Fig 5. *LINC01013* is associated with the TGF-β pathway.** A) Protein-protein interaction (PPI) network from calcified aortic valves and B) gene ontology enrichment of the proteins included in the network show enrichment for the TGF-β signaling pathway (adjusted P value).

dysregulated lncRNA-SE in CAVD. In VICs, mapping of SE revealed an enrichment for extracellular matrix organization, cell substrate adhesion and actin cytoskeleton. These functional enrichments represent salient biologic features of the AV, which are dysregulated during the

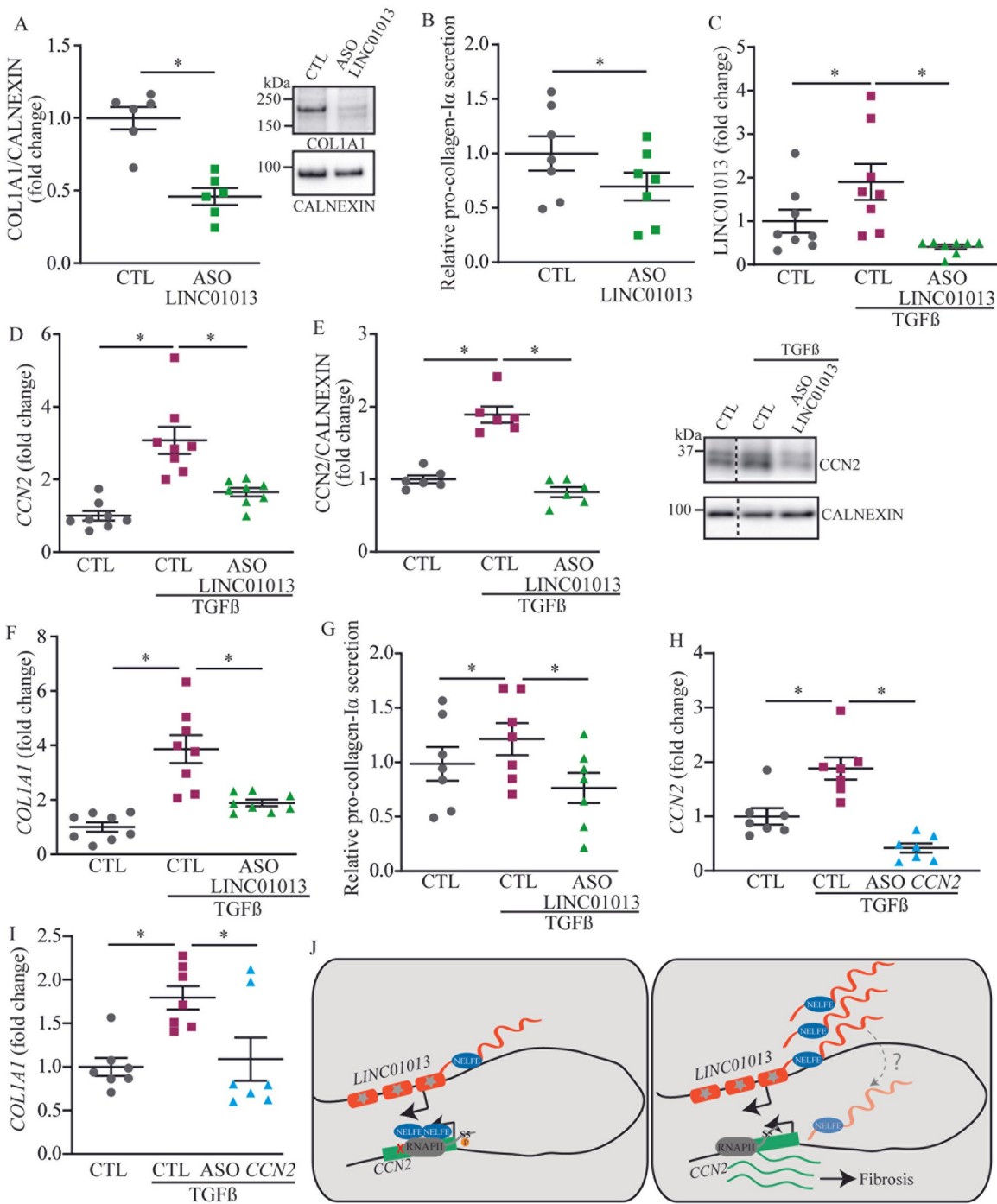

**Fig 6. TGF-β mediates *LINC01013* activation.** A-B) Western blot (n = 6) (A) and pro-collagen-Iα measurements (n = 7) (B) indicate decreased collagen following *LINC01013* ASO treatment. C) TGF-β treatment induces *LINC01013* expression (n = 8). D-E) TGF-β-mediated increase of *CCN2* mRNA expression (n = 8) (D) and protein level (n = 6) (E) rely on *LINC01013* expression. F-G) The rise in *COL1A1* expression (n = 8) and pro-collagen-Iα secretion (n = 7) following TGF-β treatment relies on *LINC01013* expression. H) TGF-β promotes *CCN2* expression (n = 7). I) *CCN2* ASO (H) negates TGF-β-induced *COL1A1* expression level (n = 7). J) Cartoon illustrates the process whereby *LINC01013* controls the expression of *CCN2*; dashed line is a hypothetical process as a direct interaction of *LINC01013* with DNA at the *CCN2* locus remains to be demonstrated.

development of CAVD. Of note, 30% of SE in VICs are lncRNA-SE and are enriched in open chromatin and loops. These findings suggested that lncRNA-SE may participate to cell fate by controlling gene expression in regulatory DNA loops. *LINC01013* was prioritized for further functional assessment as this lncRNA-SE has chromatin contacts with the promoter of *CCN2*, a gene involved in extracellular matrix organization. In explanted AVs, we found in CAVD that the promoter of *LINC01013* was enriched for H3K27ac, a mark of active chromatin. Thus, these findings highlighted that lncRNA-SE are dysregulated during CAVD and are involved into several processes pertaining to matrix organization and activation of VICs.

## *LINC01013* regulates the transcription of *CCN2*

Transcription is a highly organized process involving the recruitment of regulatory complexes along with the RNAPII [29]. Distant-acting elements modulate the transcription process through DNA loops [30]. Herein, we found that *LINC01013* is a decoy factor for NELF-E and exerts, through a long-distance looping, a control on the expression of *CCN2*. The NELF complex is known to be required for the phosphorylation on serine 5 of RNAPPII associated with RNAPII pausing [20,22]. Consistently, following the knockdown of *LINC01013*, we found that the level of RNAPII-Ser5p at the TSS of *CCN2* was increased. Hence, during CAVD, higher expression of *LINC01013* promotes the dissociation of NELF-E from chromatin and increases the transcriptional response of *CCN2*. Interestingly, the importance of *LINC01013* in the transcriptional control of *CCN2* is also observed in the endothelium. In human umbilical endothelial cells (HUVEC), the presence of SEs at the *LINC01013* locus has been also reported as well as a strong interaction between the *LINC01013* and the *CCN2* locus [31]. Also, single-cell RNA sequencing experiments in HUVEC revealed that a sub-population of cells co-expressed *LINC01013* and *CCN2* [32]. In this regard, it appears that an unexplored role for *LINC01013* on the expression of *CCN2* in endothelial cells of the aortic valve might be an additional process involved into the pathobiology of CAVD.

## TGF beta pathway

Transition of VICs towards activated cells is promoted by TGFB1. In a PPI network, CCN2 has several connections with different members of the TGF beta pathway. In line with previous works, we found in VICs that TGBFB1 induced the expression and secretion of collagen type 1, a predominant collagen within the AV. Of interest, we found that TGFB1 also induced the expression of *LINC01013*. We thus hypothesized that *LINC01013* may regulate TGFB1-dependent fibrogenesis. To this effect, ASO-mediated knockdown of *LINC01013* prevented TGFB1-induced expression of *COL1A1* and secretion of pro-collagen type 1-α. Also, TGFB1-induced expression of *COL1A1* was abrogated by the silencing of *CCN2*. Taken together, the present findings in VICs suggest that TGFB1-induced synthesis of collagen relied on a *LINC01013-CCN2* pathway. These data are consistent with the role of *CCN2* in fibrotic diseases.

## Clinical implications

The present work underlined that *LINC01013* regulates the transcription of *CCN2*, which promotes fibrogenesis, an underpinning process in the development of CAVD. Hence, these findings provide evidence that *LINC01013* could be targeted in order to control CAVD. Further work on the role of key regulators of fibrogenesis in the AV could lead to novel therapies.

## Limitations

Though we showed that *LINC01013* is a transcriptional regulator of *CCN2* by acting as decoy factor for NELF-E, the present study did not investigate the possible direct interaction of

*LINC01013* with DNA at this locus. Further experiments are needed to address the interaction of *LINC01013* with chromatin. The expression of lncRNAs was documented in surgically explanted AVs and thus it was not possible to assess whether *LINC01013* is expressed early during the development of CAVD. The role of *LINC01013* was examined *in vitro* and whether targeting its expression *in vivo* would result in lower fibrosis of the AV is presently unknown, and the development of a mouse model for translational studies could constitute a challenge: the poor conservation of *LINC01013* through species requires more investigation to identify a putative orthologue in mouse. However, conservation of synteny between human and mouse at the *CCN2* and *LINC01013* loci would suggest the presence of an orthologue, for which the sequence remains to be determined by using unbiased methods like rapid amplification of cDNA ends followed by long-read RNA sequencing (RACE-seq).

## Conclusions

We provide evidence that lnRNA-SE are dysregulated during CAVD and participate to cell fate and extracellular matrix organization. Specifically, we identified *LINC01013*, a lnRNA-SE that is spatially coordinated with *CCN2* and regulates the expression of CCN2 and the fibrotic response. Increased expression of *LINC01013* promotes the expression of CCN2 and the synthesis of collagen. Targeting *LINC01013* could prevent fibrosis of the AV.

## Methods

### Ethics statement

The protocol was approved by the ethical committee of the Institut Universitaire de Cardiologie et de Pneumologie de Québec. Written formal consent was obtained from the subjects.

### VICs isolation

VICs were isolated from non-mineralized aortic valves obtained from patients during heart transplantation procedure. Aortic leaflets cut into pieces were incubated in 0.3% type I collagenase (Invitrogen, Thermo Fisher Scientific, ON, Canada) at 37°C for 45 minutes and then filtered through a 70 μm mesh before a centrifugation for 5 minutes at 1,500 rpm. Cells were resuspended in complete media (DMEM, 10% FBS with L-glutamine and sodium pyruvate) and used between passages 3 to 7.

### ChIP-seq library preparation and sequencing

VICs isolated were fixed with 1% formaldehyde at room temperature for 10 minutes and then quenched with glycine to 0.125 M. Cells were harvested in 1 mL of PBS 1X/5% BSA (Sigma-Aldrich, ON, Canada) and then centrifuged 5 minutes at 800 x *g* at 4°C before removing the supernatant. Cells were resuspended in 1 mL of lysis buffer L1A/B (10 mM Hepes-KOH pH 7.5, 85 mM KCl, 1 mM EDTA, 0.5% NP-40 containing PMSF, sodium butyrate and PIC (Sigma-Aldrich, ON, Canada)) and incubated on ice for 10 minutes. Cellular extracts were centrifuged for 5 minutes at 8,000 x *g*, then nuclei were sonicated on ice with a Bioruptor Standard Sonicator (Diagenode, NJ, USA) in 450 μL of L3 buffer (5 mM Tris-HCl pH 7.5, 100 mM NaCl, 10 mM EDTA, 0.5 mM EGTA, 0.1% deoxycholate, 0.5% sarkosyl, supplemented with PMSF, sodium butyrate and PIC) for 3 minutes cycles alternating 10 seconds ON and 10 seconds OFF with 30% amplitude. Triton-X100 to final concentration of 1% was added and nuclear extracts were centrifuged for 5 minutes at 18,000 x *g* at 4°C. Immunoprecipitations were then performed for 2 hours at 4°C with 2 μg of H3K4me1 (#5326, Cell Signaling Technology, New England Biolabs, ON, Canada) or 2 μg of H3K27Ac (#ab4729, Abcam, ON, Canada)

pre-bound to protein G Dynabeads (Life Technologies, Thermo Fisher Scientific, ON, Canada). Samples were washed three times with ice cold wash buffer I (20 mM Tris-HCl pH 7.5, 150 mM NaCl, 2 mM EDTA, 1% Triton X-100, 0.1% SDS, 2 mM EDTA), wash buffer II (10 mM Tris-HCl, 250 mM LiCl, 1% NP-40, 0.7% deoxycholate, 1 mM EDTA) and TET (10 mM Tris-HCl pH 7.5, 1 mM EDTA, 0.1% Tween-20). DNA-proteins complex were eluted with 100 μL of 1% SDS/TE at room temperature for 40 minutes. Reverse-crosslink was performed at 65˚C overnight after addition of NaCl to 300 mM final concentration. RNA were digested with 0.33 mg/mL of RNase A (Qiagen, ON, Canada) for 1 hour at 37˚C and proteins digested for 1 hour at 55˚C with 0.5 mg/mL of proteinase K (Qiagen, ON, Canada). DNA was purified using DNA clean up & concentrator kit (Zymo Research, Cedarlane, Canada). Libraries were constructed using the QIAseq Ultralow Input Library Kit (Qiagen, ON, Canada) and PCR-amplified for 10–12 cycles. Fragments of 200–1,000 bp were selected according to manufacturer's instructions. As a control, 1% input chromatin libraries after sonication were sequenced for each ChIP samples. Sequencing was performed using 75-bp reads on an Illumina HiSeq4000 (UCSD IGM Genomics Facility, CA, USA). Data are available in GEO (accession number GSE154511).

## ATAC-seq library preparation and sequencing

Nuclei from approximatively 2.5 x $10^5$ to 4 x $10^5$ of VICs were treated as previously [33] described with some modifications. Transposition reactions were performed using the Tn5 Transposase and TD reaction buffer from the Nextera DNA library preparation kit (Illumina, CA, USA) for 30 minutes at 37˚C. The transposed DNA were then purified using DNA clean up & concentrator kit (Zymo Research, Cedarlane, Canada) and DNA were eluated in 20 μL of elution buffer (Buffer EB from DNA clean up & concentrator kit). Libraries were first amplified for 5 cycles using custom-synthesized index primers followed by a second amplification. The appropriate number of additional PCR cycles was determined using quantitative real-time PCR by plotting the Rn value (fluorescent signal from SYBR Green I) versus cycle number and determining the cycle number corresponding to one-third of the maximum fluorescent intensity. Libraries were PCR-amplified for 4–7 cycles and then purified using DNA clean up & concentrator kit (Zymo Research, Cedarlane, Canada). Library quality was assessed on a BioAnalyser (Agilent, ON, Canada) using Agilent High Sensitivity DNA kit (Agilent, ON, Canada). Sequencing was performed using 75-bp reads on an Illumina HiSeq4000 (UCSD IGM Genomics Facility, CA, USA). Data are available in GEO (accession number GSE154510).

## ChIP-seq and ATAC-seq analysis

FASTQ files were mapped to the UCSC genome build hg19 using Bowtie2 with default settings. BigWIG and peak calling were respectively generated with bamcoverage tool and MACS2 with default parameters.

## Super-enhancers analysis

Super-enhancers identification was performed as described previously [13,14] with a few modifications. First, MACS2 was used to identify enhancers based on H3K27ac peaks with a *q*-value threshold of 1 x $10^{-5}$. H3K27ac peaks which overlapped with the ENCODE [34] black-listed genomic regions were removed. The remaining H3K27ac peaks defined as enhancers were then used to identify super-enhancers (SE) by performing the ROSE [8] (Rank Ordering of Super-Enhancers) algorithm with default settings (stitching distance of 12.5 kb without TSS exclusion). GREAT [35] was used to determine Gene Ontology (GO) enrichment of SE; GREAT uses a binomial test to identify significant enrichments from the deviations of a

theoretically expected distribution of observations to control for false positives, and gives a binomial *p*-value associated with the enrichments. SE ranking by ROSE were compared with a Wilcoxon ranked-sum test. Open chromatin enrichment for SE was evaluated with the GenometricCorr [36] package. lncRNA-SE and R-SE were defined from SE by using the Bedtools intercept function with the TSS of lncRNAs from the GENCODE [15] version 32 dataset. ATAC-seq signal around lncRNA-SE and R-SE were calculated with HOMER [37] and plot with R by using a loess smoothing method. The 5kb centered ATAC-seq signal from lncRNA-SE and R-SE was compared with a Wilcoxon ranked-sum test.

## H3K27ac HiChIP

H3K27ac HiChIP data for GM12878, Naïve T cells and HCAMSC were used from GEO (accession number GSE101498) [38]. H3K27ac HiChIP for VICs was performed as described by Mumbach et al. [39]. About 1 million of VICs were crosslinked with 1% formaldehyde for 15 minutes and quenched by glycine (0.125 M final). Cells were resuspended with 500 µL of cold HiC lysis buffer (10 mM Tris-HCl pH 8.0, 10 mM NaCl, 0.2% NP-40, supplemented with PIC, Sigma-Aldrich, ON, Canada) with rotation at 4˚C for 30 minutes. Nuclei were isolated by centrifugation at 2,500 x g for 5 minutes at 4˚C, washed with cold HiC lysis buffer, resuspended with 0.5% SDS and incubated for 10 minutes at 62˚C. 285 µL of water and 50 µL of 10% Triton X-100 were then added to quench SDS followed by an incubation at 37˚C for 15 minutes. Chromatin was digested 2 h at 37˚C by addition of 40 U of MboI restriction enzyme and 50 µL of CutSmart buffer 10X (New England BioLabs, ON, Canada). Enzyme was inactivated at 62˚C for 20 minutes. To mark DNA overhangs with biotin, a mix containing biotin–dATP, dCTP, dGTP, dTTP (Thermo Fisher, ON, Canada) and DNA Polymerase I Klenow fragment (New England BioLabs, ON, Canada) was added, followed by an incubation at 37˚C for 1 hour with rotation. Ligation was then performed at room temperature for 4 hours with rotation by addition of a ligation master mix containing ligase buffer (New England BioLabs, ON, Canada), 10% Triton X-100, BSA (Thermo Fisher, ON, Canada), 4,000 U of T4 ligase (New England BioLabs, ON, Canada), and 660 µL of water. Nuclei were then isolated by centrifugation at 2,500 x g for 5 minutes and resuspended in Nuclear lysis buffer (50 mM Tris-HCl pH 7.5, 10 mM EDTA, 1% SDS, supplemented with PIC, Sigma-Aldrich, ON, Canada) before sonication on ice with a Bioruptor Standard Sonicator (Diagenode, NJ, USA). Sample was clarify for 15 minutes at 16,100 x g at 4˚C and 800 µL of ChIP Dilution buffer (0.01% SDS, 1.1% Triton X-100, 1.2 mM EDTA, 16.7 mM Tris-HCl pH 7.5, 167 mM NaCl) were added. 34 µL of protein G Dynabeads (Life Technologies, Thermo Fisher Scientific, ON, Canada) were washed and resuspended in 50 µL of ChIP Dilution buffer. Sample was then added to beads, followed by an incubation at 4˚C for 1 hour with rotation. Sample was put on a magnet and supernatant removed into a new tube. 4 µg of H3K27ac antibody (#ab4729, Abcam, ON, Canada) were added, and the sample was incubated overnight at 4˚C with rotation. 34 µL of protein G Dynabeads in 50 µL of ChIP dilution buffer were added to the sample for an incubation at 4˚C for 2 hours. Beads were then washed sequentially with Low Salt Wash buffer (0.1% SDS, 1% Triton X-100, 2 mM EDTA, 20 mM Tris-HCl pH 7.5, 150 mM NaCl), High Salt Wash buffer (0.1% SDS, 1% Triton X-100, 2 mM EDTA, 20 mM Tris-HCl pH 7.5, 500 mM NaCl) and LiCl wash buffer (10 mM Tris-HCl pH 7.5, 250 mM LiCl, 1% NP-40, 1% sodium deoxycholate, 1 mM EDTA), and resuspended in 100 µL of DNA Elution buffer (50 mM sodium bicarbonate pH 8.0, 1% SDS). Sample was incubated at room temperature for 10 minutes with rotation followed by 3 minutes at 37˚C shaking, and then placed on a magnet. The supernatant was then transferred in a new tube and another 100 µL of DNA Elution buffer was added to the beads for another same incubation to obtain 200 µL of sample. Reverse-crosslink

was performed by addition of Proteinase K (Qiagen, ON, Canada) and a first incubation at 55˚C for 45 minutes with shaking followed by a second incubation at 67˚C for 1.5 hours. DNA was then purified with clean up & concentrator kit (Zymo Research, Cedarlane, Canada) and quantified with QuBit (Thermo Fisher, ON, Canada). 5 μL of Streptavidin C-1 beads (Life Technologies, Thermo Fisher Scientific, ON, Canada) were washed with Tween Wash buffer (5 mM Tris-HCl pH 7.5, 0.5 mM EDTA, 1 M NaCl, 0.05% Tween-20) and resuspended in 10 μL of 2X Biotin Binding buffer (10 mM Tris-HCl pH 7.5, 1 mM EDTA, 2 M NaCl). Sample was then added for an incubation of 15 minutes at room temperature with rotation, and placed on a magnet. Supernatant was discarded and beads washed twice by adding 500 μL of Tween Wash buffer and incubating at 55˚C for 2 minutes shaking. Beads were washed in 100 μL of 1X TD buffer (2X TD buffer is 20 mM Tris-HCl pH 7.5, 10 mM magnesium chloride, 20% dimethylformamide), and then resuspended in 25 μL of 2X TD buffer, an appropriate amount of Tn5 adjusted for the DNA quantity (about 5 ng) and water to 50 μL, followed by an incubation at 55˚C for 10 minutes with interval shaking. Sample was then placed on a magnet and supernatant removed. 100 μL of 50 mM EDTA were then added to the sample for an incubation at 50˚C for 30 minutes, followed by two washes at 50˚C for 3 minutes, and two additional washes in Tween Wash buffer at 55˚C for 2 minutes. Beads were then resuspended in 10 mM Tris. Sample was placed on magnet and beads resuspended in 50 μL of a PCR master mix (25 μL of Phusion HF 2X (New England BioLabs, ON, Canada), 1 μL of each Nextera Ad1_noMX and Nextera Ad2.1 at 12.5 μM, and 23 μL of water). The following PCR program was performed: 72˚C for 5 min, 98˚C for 1 min, then cycle at 98˚C for 15 s, 63˚C for 30 s, and 72˚C for 1 min. Cycle number was set to 10 considering the amount of DNA. Library was placed on a magnet and eluted into a new tube before purification with clean up & concentrator kit (Zymo Research, Cedarlane, Canada). Library size selection (300–700 bp) was performed by cutting a 2% agarose gel and DNA purified with PureLink Quick Gel Extraction (Thermo Fisher, ON, Canada). Library quality was assessed on a BioAnalyser (Agilent, ON, Canada) using Agilent High Sensitivity DNA kit (Agilent, ON, Canada). Paired-end sequencing with 690 million reads was performed with NovaSeq 6000 S4 (UCSD IGM Genomics Facility, CA, USA). Data are available in GEO (accession number GSE154512).

## H3K27ac HiChIP data processing

Paired-end reads were aligned using HiC-Pro with default settings. Contact matrix were generated with Juicer from the allValidPairs file generated by HiC-Pro [40]. Juicebox [41] was used for matrix visualisation. Most confident interactions (FDR < 0.01) were determined with FitHiChIP [42]. The matrix normalized by the vanilla coverage square root for the chromosome 6 at a 5 kb resolution was dumped using Juicer. This matrix was used to generate the virtual 4C with R considering all the interactions with a bin containing the TSS of *LINC01013*. H3K27ac HiChIP 1D track for VICs was generated using bamcoverage from the BAM file generated by HiC-Pro.

## RNA-sequencing

RNA-sequencing was performed on 10 control nonmineralized AVs collected from hearts harvested during cardiac transplantation, and 19 surgically-explanted mineralized AVs (CAVD) from patients undergoing aortic valve replacement. Valve leaflets of each individual were conserved at -80˚C. One leaflet per valve was crushed with beads in TRIzol by using a Mini Bead Mill Homogenizer (VWR, QC, Canada), and RNA extracted with the RNeasy Mini Kit (Qiagen, ON, Canada) according to the manufacturer's instructions. Gene expression was evaluated by the Illumina HiSeq 2000 platform. TopHat and Cufflinks were used to align and assemble reads. Read counts matrix generated by STAR were normalized with the edgeR

package by using the calcNormFactor function with the trimmed mean of M-values (TMM) method. Differential gene expression between groups of valves were then compared with Limma. Correction for multiple testing was performed with Benjamini and Hochberg correction. LncRNAs with an adjusted value of P<0.05 were considered differentially regulated.

### Chromosome conformation capture (3C)

3C experiments were performed as described previously [43]. Briefly, 10 millions of VICs were crosslinked with 1% formaldehyde for 15 minutes, quenched by glycine (0.125 M final), and lysed with cold lysis buffer (10 mM Tris-HCl pH 7.5, 10 mM NaCl, 0.2% NP-40, supplemented with PIC, Sigma-Aldrich, ON, Canada). Nuclei were isolated by centrifugation at 400 x $g$ for 5 minutes at 4˚C, and then resuspended in CutSmart buffer 1.2X (New England BioLabs, ON, Canada) plus 0.3% SDS. Nuclei were incubated 1 hour at 37˚C while shaking at 900 rpm before the addition of Triton X-100 (2% final) and incubated another hour with the same conditions. DNA were digested overnight at 37˚C with 400 U of PstI (New England BioLabs, ON, Canada), and the enzyme was then inactivated by addition of 1.6% SDS at 65˚C for 25 minutes. Reactions were then diluted for 20-fold by ligation buffer 1.15X (New England BioLabs, ON, Canada) with 1% Triton X-100, and ligated with 1,000 U of T4 ligase (New England BioLabs, ON, Canada) at 16˚C overnight. Reverse crosslink was then performed by Proteinase K (Qiagen, ON, Canada) addition at 65˚C overnight. RNA was digested with RNase A (Qiagen, ON, Canada) and ligated DNA fragments were then purified by phenol/chloroform extraction, and analyzed by quantitative real-time PCR (qPCR).

Quantification of ligated products was performed by qPCR using the PstI cutting site located upstream the TSS of *CCN2* as the anchor. For all fragments, we first normalized using a genomic *GAPDH* undigested control, then we used the level of ligation products between the anchor and the nearest downstream PstI site (#1) as a strong normalizer to correct for genomic random interactions as described previously [44]. Two biological replicates were conducted and qPCR were performed in triplicates to improve sensitivity. Primers sequences are provided in the S6 Table.

### Cell transfection and TGF beta treatment

VICs were transfected with HiPerFect transfection reagent (Qiagen, ON, Canada). Knockdown of *LINC01013* and *CCN2* were performed 72 hours with a transfection of an oligonucleotide antisens (ASO) against *LINC01013* (80 nM), *CCN2* (40 nM) or NC5 as a negative control (IDT, IL, USA). TGF beta stimulations were performed 48 hours after 24 hours of transfection using 10 ng/mL of recombinant human TGF beta 1 (Life technologies, Thermo Fisher Scientific, ON, Canada).

### ChIP-qPCR

Mineralized and non-mineralized tissues from aortic valve were homogenized in 1 ml of phosphate-buffered saline (PBS) 1X containing PIC (Sigma-Aldrich, ON, Canada). The homogenized samples were centrifuged 5 minutes at 800 x $g$ at 4˚C and pellets were resuspended in PBS 1X containing PIC and centrifuged 5 min at 800 x $g$ at 4˚C. Pellets were then used to perform ChIP experiment. ChIP in VICs were performed with about 2 millions of cells. Pellet from tissues or VICs were crosslinked with 1% formaldehyde for 15 minutes, quenched by glycine (0.125 M final) and centrifuged 5 minutes at 800 x $g$ at 4˚C. The supernatants were removed and pellets were then resuspended in 500 μL of lysis buffer (50 mM Hepes-KOH pH 7.5, 140 mM NaCl, 1 mM EDTA, 1% Triton X-100, 0.1% sodium deoxycholate, 0.1% SDS, supplemented with PIC, Sigma-Aldrich, ON, Canada). Sonication led to fragmentation of an

average length of 400 bp. 2 μg of antibodies against H3K27Ac (#ab4729, Abcam, ON, Canada), RNAPII (#ab26721, Abcam, ON, Canada), RNAPII-Ser5p (#ab5131, Abcam, ON, Canada), NELFE (#10705-1-AP, Proteintech, IL, USA) or isotype IgG (Cell Signaling Technology, New England Biolabs, ON, Canada) were incubated with proteins G dynabeads (Life technologies, Thermo Fisher Scientific, ON, Canada) for 6 hours before DNA samples were added to the antibodies/dynabeads mixtures and incubated at 4°C overnight on a rotator. The next day, samples were washed with low salt buffer (0.1% SDS, 1% Triton X-100, 2 mM EDTA, 20 mM Tris-HCl pH 8.0, and 150 mM NaCl). A second washing with high salt buffer (0.1% SDS, 1% Triton X-100, 2 mM EDTA, 20 mM Tris-HCl pH 8, and 500 mM NaCl) was performed, followed with LiCl buffer (10 mM Tris-HCl pH 8, 250 mM LiCl, 1 mM EDTA, 1% NP-40) and finally with TE buffer (10 mM Tris-HCl, pH 8, 1 mM EDTA). The complex was then eluted by adding 200 μL of elution buffer (1% SDS, 100 mM NaHCO$_3$) at 65°C for 1 hour. Eluted samples were reverse crosslinked at 65°C overnight. DNA fragments were purified using DNA clean up & concentrator kit (Zymo Research, Cedarlane, Canada) and samples were analysed by qPCR assay using primers specific to the TSS region of *LINC01013* (F:gaggcagtttcctatctggttt, R:tctatccagaccacttgtgtttag), *CCN2* (F:actggctgtctcctctcagc, R:tgtaggactccattcagctcat).

## Nuclear and cytoplasmic RNA purification

Nuclear and cytoplasmic quantification of *LINC01013* were performed with the Cytoplasmic and Nuclear RNA Purification Kit (Norgen Biotek, ON, Canada) according to the manufacturer's instructions. *U6* and *HPRT* were used as controls respectively for the nuclear and cytoplasmic compartments. We normalized data for each compartment with the total RNA level in cells.

## Quantitative real-time PCR

RNA from mineralized and non-mineralized aortic valve tissues was purified by using the RNeasy Mini kit (Qiagen, ON, Canada). RNA from VICs was isolated with E.Z.N.A. Micro RNA kit (Omega Bio-tek, VWR, QC, Canada). One μg of RNA was reverse transcribed using the Qscript cDNA supermix from Quanta (VWR, QC, Canada). qPCR were performed with perfecta sybr supermix from Quanta on the Rotor-Gene 6000 system (Corbett Robotics Inc, CA, USA). Primers for *LINC01013* were obtained from IDT (IDT, IL, USA) (F:ctaaacacaagtggtctggataga, R:tttggatcaccaaggcaag). Primers for *CCN2* and *COL1A1* were purchased from Qiagen (ON, Canada). The expression of the hypoxanthine-guanine phosphoribosyltransferase (*HPRT*) (Qiagen, ON, Canada) was used as a reference to normalize the results.

## Western blotting

Cell extracts from VICs were boiled for 10 minutes and proteins were loaded onto polyacrylamide gels followed by electrophoresis and transferred to nitrocellulose membranes. Membranes were blocked with TBS 1X/0.1% Tween-20 containing 5% non-fat dry milk (Sigma-Aldrich, ON, Canada), incubated with antibodies against CCN2 (#E-AB-12339, Elabscience, TX, USA), NELF-E (#10705-1-AP, Proteintech, IL, USA), COL1A1 (#84336, Cell Signaling Technology, New England Biolabs, ON, Canada) or CALNEXIN (#2679, Cell Signaling Technology, New England Biolabs, ON, Canada) overnight at 4°C. Membranes were then washed and incubated with HRP-labeled secondary antibodies (TransBionovo Co., Civic Biosciences Ltd, QC, Canada). Detection was done using clarity western ECL substrate (BioRad, ON, Canada). Images were acquired and quantification analyses were performed using a ChemiDocMP (BioRad, ON, Canada) system and the ImageJ software.

## ELISA

The pro-collagen-Iα excretion by VICs was quantified from 100 µL of VICs culture medium using the Human Pro-Collagen I alpha 1 DuoSet ELISA kit (R&D systems, MN, USA) according to the manufacturer's instructions.

## RIP

About 5 millions of VICs were used to perform RIP analysis. First, 20 µg of antibodies against NELF-E (Proteintech, IL, USA) or isotype IgG (Cell Signaling Technology, New England Biolabs, ON, Canada) were incubated with 50 µL of proteins G dynabeads (Life technologies, Thermo Fisher Scientific, ON, Canada) in 500 µL of IP buffer (0.5% Triton X-100, 200 mM NaCl, 10 mM Tris-HCl at pH 7.5 and 10 mM EDTA) at 4˚C overnight on a rotator. Cells were resuspended in cold lysis buffer (0.5% Triton X-100, 10 mM NaCl, 10 mM Tris-HCl pH 7.5, 10 mM EDTA, 0.5 mM PMSF, 1 mM DTT, supplemented with PIC and ProtectRNA, Sigma-Aldrich, ON, Canada) and incubated 20 minutes on ice before a centrifugation at 400 x $g$ for 10 minutes at 4˚C. NaCl was added to a final concentration of 200 mM and lysate was pre-cleared with 50 µL of proteins G dynabeads in IP buffer 1 hour at 4˚C on a rotator. The cleared lysate was then transferred to the tubes containing NELF-E of IgG-coated beads for 3h at 4˚C on a rotator. Samples were washed five times with 500 µL of IP buffer and then resuspended in RTK lysis buffer from E-Z Nucleic Acid (E.Z.N.A.) kit (Omega Bio-tek, VWR, QC, Canada). RNA was reverse transcribed using the Qscript cDNA supermix from Quanta (VWR, QC, Canada). qPCR were performed with perfecta sybr supermix from Quanta on the Rotor-Gene 6000 system (Corbett Robotics Inc, CA, USA).

## Network analysis

Based on TissueNet [45] data, an experimentally validated dataset, protein-protein interaction (PPI) pairs were extracted in order to generate a PPI network. From data generated with the RNAs-seq in explanted mineralized AVs (TPM higher than the 10[th] percentile), PPI pairs were inferred based on the TissueNet data. To generate the PPI network, the interactome of CCN2 including COL1A1, SMAD2, SMAD3 and TFGB1 was extracted. To that end, interaction pairs which included CCN2, COL1A1, SMAD2, SMAD3 and TFGB1 were isolated to form an edge list. To visualize the network, NetworkAnalyst [46] was used.

## Collagen immunostaining

Human VICs were seeded on poly-L-lysine coated glass coverslips. Cells were washed once with PBS 1X and fixed in 3.7% formaldehyde for 30 minutes at room temperature. Cells were treated 20 minutes with 50 mM NH4Cl in PBS 1X and permeabilized for 10 minutes in PBS 1X containing 0.2% Triton X-100. Cells were then incubated in PBS 1X containing 5% milk for one hour at room temperature with constant agitation. Incubation with anti-COL1A1 (#E8F4L, Cell Signaling Technology, New England Biolabs, ON, Canada) was performed in PBS 1X containing 1% milk overnight at 4˚C. Cells were washed four times with PBS 1X, followed with one hour incubation with FITC-conjugated anti-rabbit secondary antibody (Molecular Probes, Thermo Fisher Scientific, ON, Canada). Secondary antibody alone was used as negative control. Slides were mounted and analysed images were acquired using a Zeiss microscope LSM800 driven by the Zen software (Objective 20X oil, 1.4 NA, Zeiss, ON, Canada). Image processing and quantification were performed with ImageJ 1.47g (NIH, USA). Integrated densities for each cell were calculated after removal of the background for the negative control.

## Statistical analysis

Continuous data were expressed as mean ± SEM. Normality was tested with the Shapiro-Wilk test. For two groups, data with normal distribution were compared with Student t-test, for more than two groups with ANOVA. For data with non-normal distribution, two groups were compared with the Wilcoxon-Mann-Whitney test or with the Kruskal-Wallis test for more than two groups. A multivariate regression was used to adjust for age and sex. Categorical data were compared with Fischer's exact test. Statistical analysis were performed with JMP 14.2.0 or Prism 8.0.2.

## Supporting information

**S1 Fig. Experimental approaches used from aortic valve tissues and valvular interstitial cells (VICs) for multidimensionnal profiling and functionnal analysis.**
(TIFF)

**S2 Fig. H3K27ac HiChIP contact matrices in VICs.**
(TIFF)

**S3 Fig. Scheme of experiments and analysis conducted for the prioritization of dysregulated lncRNA-SEs in CAVD.**
(TIFF)

**S4 Fig.** A) ChIP-qPCR of NELF-E at the promoter of LINC01013. B) COL1A1 immunostaining on VICs. Conditions with NC5 or ASO against LINC01013 were compared with a paired t-test, and the TGFB conditions with ANOVA and Tukey's test for post-hoc analysis. n = 50 for each condition.
(TIFF)

**S1 Table. Super-enhancers identified in VICs with the ROSE algorithm.**
(XLSX)

**S2 Table. Super-enhancers identified in VICs with the HOMER algorithm.**
(XLSX)

**S3 Table. List of the lncRNA-SE.**
(XLSX)

**S4 Table. lncRNAs differentially expressed between the control nonmineralized and mineralized aortic valves.**
(XLSX)

**S5 Table. lncRNAs differentially regulated in CAVD derived from SE-lncRNAs.**
(XLSX)

**S6 Table. Protein coding genes interacting with lncRNAs differentially regulated derived from SE-lncRNAs.**
(XLSX)

**S7 Table. Primer sequences for the 3C experiment.**
(XLSX)

**S8 Table. Coding potential of LINC01013 with CPAT.**
(XLSX)

## Author Contributions

**Conceptualization:** Arnaud Chignon, Marie-Chloé Boulanger, Valentin Bon-Baret, Sébastien Thériault, Yohan Bossé, Patrick Mathieu.

**Data curation:** Arnaud Chignon.

**Formal analysis:** Arnaud Chignon, Zhonglin Li, Patrick Mathieu.

**Funding acquisition:** Patrick Mathieu.

**Investigation:** Arnaud Chignon, Déborah Argaud, Marie-Chloé Boulanger, Ghada Mkannez, Valentin Bon-Baret, Patrick Mathieu.

**Methodology:** Arnaud Chignon, Déborah Argaud, Patrick Mathieu.

**Project administration:** Patrick Mathieu.

**Software:** Zhonglin Li.

**Supervision:** Patrick Mathieu.

**Validation:** Patrick Mathieu.

**Visualization:** Marie-Chloé Boulanger.

**Writing – original draft:** Arnaud Chignon, Patrick Mathieu.

**Writing – review & editing:** Déborah Argaud, Marie-Chloé Boulanger, Ghada Mkannez, Valentin Bon-Baret, Zhonglin Li, Sébastien Thériault, Yohan Bossé.

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
