## [Decision Letter · Decision Letter 0]

18 May 2021

Dear Dr Mathieu,

Thank you very much for submitting your Research Article entitled 'Genome-Wide Chromatin Contacts of Super-Enhancer-Associated lncRNA Identify RECAST as a Regulator of Fibrosis in the Aortic Valve' to PLOS Genetics.

The manuscript was fully evaluated at the editorial level and by independent peer reviewers. The reviewers appreciated the attention to an important problem, but raised some substantial concerns about the current manuscript. Based on the reviews, we will not be able to accept this version of the manuscript, but we would be willing to review a much-revised version. We cannot, of course, promise publication at that time.

If you decide to revise the manuscript for further consideration at PLOS Genetics, please aim to resubmit within the next 60 days, unless it will take extra time to address the concerns of the reviewers, in which case we would appreciate an expected resubmission date by email to plosgenetics@plos.org.

[LINK]

We are sorry that we cannot be more positive about your manuscript at this stage. Please do not hesitate to contact us if you have any concerns or questions.

Yours sincerely,

Jeannie T. Lee

Consulting Editor - PLoS Genetics

PLOS Genetics

John Greally

Section Editor: Epigenetics

PLOS Genetics

Reviewer's Responses to Questions

**Comments to the Authors:**

Reviewer #1: In their manuscript, Chignon et al. seek to determine if super enhancer (SE)-derived lncRNA RECAST regulates fibrosis in the aortic valve. Their study uses an impressive array of molecular biology techniques, next-generation sequencing, and bioinformatic analyses to identify a putative SE-derived lncRNA associated with calcific aortic valve disease. They identified RECAST as an SE-lncRNA that positively controls CCN2 expression by acting as a decoy for the NELF-E complex, leading to fibrosis. Although of interest, the manuscript should be edited for clarity and to provide sufficient background to lead the reader logically through the experiments. In addition, the studies supporting RECAST sequestration of NELF_E should be linked to functional studies.

Main comments

1) In the introduction, lncRNAs are presented as fine-tuners of gene expression and as such could be targeted to reduce fibrosis. It would be appropriate to introduce the successes or failure of conventional therapies targeting fibrotic pathways in cardiovascular disease before discussing a potentially new therapeutic target. As one of the major findings concerns CCN2, it would be helpful to introduce CCN2 in the introduction as well.

2) The two different models used in the manuscript (VIC, AV from patients) need to be clearly introduced and it should be noted which model data comes from. As written, it is often unclear which of the two models the data comes from.

3) A clear statement of the objective/hypothesis is needed in the first paragraph of the result section, in order to help understand the progression of the experiments pertaining to Figure 1. For example, what is used as a control for the enrichment in SE in VICs is shown in Figure 1? If an enrichment is observed, what is base value for this enrichment?

4) Provide context for using the ROSE algorithm. Is it to identify the SE or to rank the SE? Can the findings be replicated using other algorithms such as MARGE or ChromHMM?

5) Figure 1:

a. Why switch from Box Plot to Violin Plot? Is Figure 1E a different representation of Fig 1F?

b. R-SEs are introduced in Figure 1, what is it besides a ‘regular SE’?

6) In Figure 2, why present data from chromosome 6? It only becomes clear later that CCN2 and RECAST are located on chr.6. My general issue with this sequence of figures/data is that it is presented as an unbiased systemic approach, but the data clearly suggests that the authors had already identified RECAST and are using the experiments presented in Figure 1 and 2 to justify the investigation. The authors should be transparent about the sequence of experiment leading to the discovery of RECAST.

7) Figure 2B: It is unclear how the AV were prepared for RNA-sequencing. In addition, the authors report changes in lncRNA expression, but what of mRNA expression genes implicated in the fibrotic processes of interest to the study (CCN2, SMAD2/3, and TGFB1). Along those lines, Figure 2C is not-informative as it’s unreadable.

8) The authors claim, using Figure 2D and 3A, that the locus of RECAST interacts with the promoter region of CCN2, 180kb upstream. However, the HiC data provided shows many other possible connections. This should be discussed.

9) This reviewer is confused about Figure 3B.

10) In Figure 3C, what is the rational for performing an adjustment for sex and age? Is there a sex and age component to RECAST expression? If so, please report.

a. In addition, is table 1 a supplemental table or will it be included in the manuscript?

b. Please report the statistical analysis used to perform the adjustment.

11) Figure 3D, are the IgG data matched per sample or were a defined set of samples chosen for the IgG vs the H3K27ac procedure? Why are there fewer samples in the control H3K27ac compared to other conditions?

12) The authors should perform RNA-FISH in combination with IF, as well as DNA-FISH in combination with RNA-FISH to show colocalization of CCN2, RECAST, RECAST-SE-locus and NEFLE.

13) Figure 4A should include appropriate controls for cytoplasmic and nuclear RNAs.

14) Figures 4 and 6 would benefit from either functional data and/or collagen immuno-staining to show effects of the RECAST ASO on whole cells, not just cell lysate.

15) More mechanistic data is needed to show that NELF-E is sequestered from the promoter of CCN2 by RECAST. If the model presented in Figure 6J is true, ChIP of NELF-E in cells treated with the RECAST ASO should reveal stronger attachment of NELF-E to the promoter region of CCN2, compared to control cells where RECAST sequesters NELF-E.

Reviewer #2: 1. A Google Scholar search for LINC01013 shows there are already several manuscripts about this lncRNA, including one that gave it a different name, Aerrie (https://www.ncbi.nlm.nih.gov/pmc/articles/PMC7829583/ ). The results of the new study should be discussed in the context of the previous research, and it is better to use the name already given in the literature (and to make sure its registered at genenames.org).

2. Is RECAST conserved in mouse? Is the SE it is transcribed from conserved in mouse? If yes – is it active in similar cells? These questions are important for future studies of RECAST in vivo and should be discussed.

3. Does knockdown of RECAST affect levels of NELF-E recruited to the SE in the RECAST promoter region?

4. Since a main argument of the manuscript is that RECAST affects Pol2 pausing at the CCN2 promoter, the authors should use ChIP-seq or Cut&Run to measure Pol2 pausing in the control and the ASO-treated cells. Cut&Run does not require a large number of cells, so it should be feasible. This will allow the authors to measure pausing directly, and will provide much more convincing evidence than the presented qPCR data on no change in Pol2 levels and increased in Ser5P-modified Pol2.

Minor comments:

1. Abstract “located at ~180 kb” – its not clear what this refers to - should be “at ~180 kb upstread/downstream of CCN2”.

2. Figure 1A – many of the categories shown are very similar/redundant. Can the authors filter out categories that overlap by >X% of the genes?

3. “R-SE” should be defined in the legend of Fig. 1C

4. Fig. 2A – the authors should show the positions of RECAST and CCN2 also on the vertical axis. Also, it should be indicated in the legend that these are HiChIP data (as this presentation is typically used for HiC).

5. Its not clear what Fig. 3B is showing – just the position of the primers? Where is the evidence on which isoform is expressed? Anyhow, this should be a supplementary panel.

6. Fig. 4A – the data used for normalization (cytoplasmic/nuclear RNAs) should be shown for scale.

7. Page 5 “1085 SE (Suppl. Table 1), which were highly enriched in gene ontology” – its is not clear how SEs can be enriched in gene ontology. Where these SEs first assigned to genes and the genes were tested for enrichment? How many genes? How were these assigned? This should be clarified in the results section.

8. Page 6 “We identified 324 lncRNA-SE” – does this refer to the number of lncRNAs or the number of SEs? Assuming a SE can overlap more than 1 lncRNA and perhaps vice versa, this should be clarified.

**Have all data underlying the figures and results presented in the manuscript been provided?**

Reviewer #1: Yes

Reviewer #2: Yes

PLOS authors have the option to publish the peer review history of their article (what does this mean?). If published, this will include your full peer review and any attached files.

Reviewer #1: No

Reviewer #2: No

---

## [Decision Letter · Decision Letter 1]

15 Oct 2021

Dear Dr Mathieu,

Thank you very much for submitting your Research Article entitled 'Genome-Wide Chromatin Contacts of Super-Enhancer-Associated lncRNA Identify LINC01013 as a Regulator of Fibrosis in the Aortic Valve' to PLOS Genetics.

The manuscript was fully evaluated at the editorial level and by independent peer reviewers. The reviewers appreciated the attention to an important topic but identified some concerns that we ask you address in a revised manuscript

We therefore ask you to modify the manuscript according to the review recommendations. Your revisions should address the specific points made by each reviewer.

[LINK]

Yours sincerely,

Jeannie T. Lee

Consulting Editor - PLoS Genetics

PLOS Genetics

John Greally

Section Editor: Epigenetics

PLOS Genetics

Reviewer's Responses to Questions

**Comments to the Authors:**

Reviewer #1: The authors of ‘Genome-Wide Chromatin Contacts of Super-EnhancerAssociated lncRNA Identify LINC01013 as a Regulator of Fibrosis in the Aortic Valve’ have successfully responded to many of the issues brought up by the reviewers. They have improved their manuscript for clarity and (scientific) depth. For example, the addition of secondary (in silico) approaches, such as HOMER, extra supporting information, more detailed methodologies and further explanation how certain enrichment analysis are used have boosted the overall quality of the paper.

There is one issue that still remains: In the graphical representation of the proposed molecular mechanism, the authors propose an aggregate of Linc01013, NELF-E, RNAPII and CCN2 upon treatment with TGFbeta. This putative interaction is, as the authors describe in their rebuttal, shown by HiChIP and 3C. Indeed, HiChIP can be used to determine protein-directe genome architecture and 3C to capture the 3D conformation of the chromosome. However, neither of these techniques (as far as this reviewer is aware) can pinpoint RNA presence at the DNA level. However, it would be of high relevance if the authors can show the interaction on a more granular level and the RNA-transcripts is present at this particular genomic location. As the authors claim that the implementation of DNA-FISH is labor-intensive

this can be shown in a variety of other ways:

- RNA-FISH for Linc01013 + IF for NELF-E to show colocalization of the two molecules in the cell.

- ChIRP-PCR to show localization of Linc01013 to the genomic locus presented in Figure 6J (ChIRP-Seq to see where else NELF-E and Linc01013 could work together would be best, but likely out of scope of this project).

- Enhance pull-down data to further validate the interaction between NELF-E and Linc01013 with in silico evidence (catRAPID or alike), followed by (simple) mutation in vitro studies on the RNA/protein interaction domain (overexpression followed by RIP-PCR).

Reviewer #2: The authors have addressed my comments from the previous round of review in a largely satisfactory manner, and have improved the manuscript. They should nevertheless, discuss their results in the context of the other studies on LINC01013, and mention the fact that HiCHIP data are presented in Fig. 2A in the figure legend, both points as requested in the previous round of review. I can then recommend publication in PLoS Genetics.

**Have all data underlying the figures and results presented in the manuscript been provided?**

Reviewer #1: Yes

Reviewer #2: Yes

PLOS authors have the option to publish the peer review history of their article (what does this mean?). If published, this will include your full peer review and any attached files.

Reviewer #1: No

Reviewer #2: No

---

## [Editor Report · Decision Letter 2]

24 Nov 2021

Dear Dr Mathieu,

Thank you very much for submitting your Research Article entitled 'Genome-Wide Chromatin Contacts of Super-Enhancer-Associated lncRNA Identify LINC01013 as a Regulator of Fibrosis in the Aortic Valve' to PLOS Genetics.

The manuscript was fully evaluated at the editorial level and by independent peer reviewers. The reviewers appreciated the attention to an important topic and felt that your manuscript is much improved.  They have a few remaining (minor) concerns that we hope you would address in a revised manuscript. Your revisions should address the specific points made by each reviewer.

[LINK]

Yours sincerely,

Jeannie T. Lee

Consulting Editor - PLoS Genetics

PLOS Genetics

John Greally

Section Editor: Epigenetics

PLOS Genetics

---

## [Editor Report · Decision Letter 3]

22 Dec 2021

Dear Dr Mathieu,

We are pleased to inform you that your manuscript entitled "Genome-Wide Chromatin Contacts of Super-Enhancer-Associated lncRNA Identify LINC01013 as a Regulator of Fibrosis in the Aortic Valve" has been editorially accepted for publication in PLOS Genetics. Congratulations!

Yours sincerely,

Jeannie T. Lee

Consulting Editor - PLoS Genetics

PLOS Genetics

John Greally

Section Editor: Epigenetics

PLOS Genetics

Comments from the reviewers (if applicable):

**Data Deposition**

http://datadryad.org/submit?journalID=pgenetics&manu=PGENETICS-D-21-00352R3

**Press Queries**

---

## [Editor Report · Acceptance letter]

11 Jan 2022

PGENETICS-D-21-00352R3 

Genome-Wide Chromatin Contacts of Super-Enhancer-Associated lncRNA Identify LINC01013 as a Regulator of Fibrosis in the Aortic Valve 

Dear Dr Mathieu, 

We are pleased to inform you that your manuscript entitled "Genome-Wide Chromatin Contacts of Super-Enhancer-Associated lncRNA Identify LINC01013 as a Regulator of Fibrosis in the Aortic Valve" has been formally accepted for publication in PLOS Genetics! Your manuscript is now with our production department and you will be notified of the publication date in due course.

With kind regards,

Anita Estes

PLOS Genetics

On behalf of:
